# Identification of urban land use efficiency by indicator-SDG 11.3.1

**Guoyin Cai** [1,2]*, **Jinxi Zhang**[1], **Mingyi Du**[1,2], **Chaopeng Li**[1], **Shu Peng**[3]

**1** School of Geomatics and Urban Spatial Informatics, Beijing University of Civil Engineering and Architecture, Beijing, China, **2** Beijing Advanced Innovation Center for Future Urban Design, Beijing University of Civil Engineering and Architecture, Beijing, China, **3** National Geomatics Center of China, Beijing, China

* cgyin@bucea.edu.cn

## Abstract

Inefficiency in urban land use is one of the problems caused by rapid urbanization. The UN Sustainable Development Goals (SDGs) indicator 11.3.1 is designed to test urban land use efficiency. This study employed geospatial and statistical data to compute land use efficiencies from 1990 to 2015 with five 5-year and ten 15-year intervals in Wukang, center of Deqing County, China. A flowchart was designed to extract the built-up lands from multiple data sources. The produced built-up lands were demonstrated to provide good accuracy by constructing an error matrix between the extracted and manually interpreted built-up lands as classified and reference images, respectively. By using the model provided by UN metadata to calculate SDG 11.3.1, the land use efficiencies from 1990 to 2015 were identified in Wukang. Our results indicate that the land use efficiency in Deqing County center is lower than the average of cities around the world, primarily because our in-situ study focused on a county center with larger rural regions than urban areas. Over the long term, urban land use becomes denser as the population grows, which will have a positive impact on the sustainability of urban development. This work is helpful for the local government to balance urban land consumption and population growth.

## Introduction

Environmental degradation and social tensions have put societies under severe pressures in the past couple of decades, the growing populations and increasing per capita land consumption are two key drivers [1–4]. With the rapid development of urbanization, a large proportion of cities have high consumption of suburban green spaces [5]. As stated in the World Urbanization Prospects [6], "the proportion of the world's population living in urban areas is expected to increase, reaching 66% by 2050." As a result, the global land rush—the phenomenon of domestic and transnational companies, governments, and individuals buying or leasing large tracts of farm lands, green spaces, or other water-pervious surfaces—has emerged as a critical issue, especially in Africa, Asia, and Latin America, over the last decades [7–9]. A project jointly completed by the African Union, African Development Bank, and Economic

**Data Availability Statement:** All relevant data are within the manuscript and its Supporting information files.

**Funding:** This work is financially supported by National Natural Science Foundation (NSFC) (Key

Project #41390650), the Land Use Change Detection from Satellite GF-7 by Ministry of Housing and Urban-Rural Development of the People's Republic of China (No. 06-Y20A17-9001-17/18), and by The Fundamental Research Funds for Beijing University of Civil Engineering and Architecture (No. 2018N060301). The funders had no role in study design, data collection and analysis, decision to publish, or preparation of the manuscript.

**Competing interests:** The authors have declared that no competing interests exist.

Commission on 120 cities showed that urban land cover has grown more than three times as much as the urban population. This rate of urban expansion has hindered sustainable urban development [10]. The United Nations Sustainable Development Summit held in September, 2015 adopted a new framework titled "Transforming Our World: The 2030 Agenda for Sustainable Development," which is used to guide the direction of social, economic, and environmental development from 2015 to 2030. The agenda includes 17 sustainable development goals (SDGs), 169 indicators, and 232 indices covering the above three aspects. [11]. These 17 SDGs concern challenges related to poverty, inequality, climate, environmental degradation, prosperity, disaster management, and reducing economic inequality. These factors are interrelated, and the key to achieving one goal often depends on the solution of other goal-related issues. SDG 11, specific for sustainable urban development, is to "make cities and human settlements inclusive, safe, resilient, and sustainable" [12]. Other than the disorderly expansion of cities as mentioned above, rapid urbanization has brought many problems and challenges, including the increasing number of slum dwellers [13, 14], increased air pollution [15–17], large volumes of solid waste [18], and insufficient or unaffordable basic services and infrastructure [19], which have made cities more vulnerable to disasters. There are 10 targets and 15 indicators in SDG 11 that address the challenges associated with rapid urbanization and promote the adoption of policies to make cities inclusive, resilient, safe, and sustainable, while these challenges are considerable in many middle-income countries. In this work, we focused on indicator SDG 11.3.1, the ratio (simply LCRPGR) of the land consumption rate (LCR) to population growth (PGR). It was designed to identify land use efficiency [20, 21].

Land use efficiency (LUE) refers to an increase in the output of a unit land area related to regional social and economic activities [22, 23]. LUE is a representative concept adhering to the sustainable development paradigm [24, 25] and is the result of dynamic processes driven by economic, social, traffic, and political factors [26]. Many indices, such as development density, population density, employment density, investment intensity, and economic output per land unit have been employed to measure the LUE in many previous studies [27–30]. With rapid urbanization, larger numbers of people move to urban areas, and city boundaries typically expand to accommodate the new inhabitants. The indicator SDG 11.3.1 is designed to understand the relationship between population shifts and urban land. The results from SDG 11.3.1 can help policymakers and planners ensure that cities remain economically productive and environmentally sustainable (https://unstats.un.org/sdgs/metadata).

In terms of a comprehensive scoping review of 23 existing databases, such as the UN Global SDG Indicators Database, World Bank SDG monitoring system, UrbanLex, Open Government Partnership, and ICLEI Local Governments for Sustainability, Rozhenkova et al. (2019) found that the current existing databases are insufficient for the purposes of large-scale comparative analysis because of the significant gap in policy data, especially at the city-level, and the non-system compilation of data covering the SDG 11 target [31]. Therefore, SDG 11.3.1 was categorized as insufficient information available for geospatial analysis (https://unstats.un.org/sdgs/iaeg-sdgs/tier-classification/). The Global Human Settlement Layer (GHSL) supplies open data on the spatial distribution of the built-up areas, population, and settlement typologies for the corresponding years of 1990, 2000, and 2015. These data were produced from the best open data available to map built-up areas extracted from Landsat imagery and Gridded Population of the World (GPW), v4 population estimates (http://sedac.ciesin.columbia.edu/data/collection/gpw-v4). The potential and application of the global built-up area and global population densities from the GHSL project have been discussed and implemented in monitoring the progress in Agenda 2030 SDG 11.3.1 [32]. The results calculated in the SDG 11.3.1 have been used to monitor land use efficiency by using the national official statistics of the population in Sweden, the NASA GEOSTAT grid cluster data, and GHSL [33]. Their results

indicated that open and easily accessible geospatial data can support monitoring in many aspects of development. The global built-up areas and population densities could be used directly in the development of the indicator SDG 11.3.1. However, some settlements could not be detected due to their size or construction materials or because they were below a dense tree canopy, causing some false detections from the GHSL built-up area product. Compared with open data sources, satellite images, such as Thematic Mapper (TM), Enhanced Thematic Mapper (ETM+), and Operational Land Imager (OLI) from Landsat satellites, the global land cover dataset (Globeland30) with a 30-m spatial resolution, and the urban footprint can play a significant role in a variety of geographical studies, in which sustainable urban development is a key application field [34].

In contrast to accessible global open data, localization is a substantial problem of SDG indicators [35]. On the one hand, global datasets often have a coarser resolution, which makes them unsuitable for local-level studies. However, the specific application of the SDG goal is calculated by diverse actors in widely different cities. Many studies have focused on the calculation of indicator SDG 11.3.1 in big cities on a regional or global scale. Few studies have focused on county-level cities. The objective of this work is to provide an approach for localizing SDG 11.3.1 to examine land use efficiency at the county level by combining spatial and census data. The remainder of this paper is organized as follows: Section 2 introduces the methodology including the study area, data sources, and data processing, and the method to calculate land use efficiency. Section 3 details the results and analysis, Section 4 provides a discussion, and Section 5 concludes the paper.

## Materials and methods

### Study area

Deqing is a county affiliated with Huzhou, Zhejiang province, China. It belongs to the subtropical humid monsoon climate, with four clear seasons and an annual average temperature of 17.2˚C. Deqing County is located north of Hangzhou and west of Shanghai, which makes Deqing an important node in the proximity of the cities. The county consists of 11 townships with a household registered population of 430,000. The land cover in Deqing is characterized by 50% mountains, 10% water, and 40% farmland, forming a livable place with mountain and water views [36]. Based on the statistics in 2006, the total land use area in Deqing County was 93,792.82 ha, with agricultural land, including cultivated land, garden land, forest land, and other agricultural land, accounting for 77.59%. Construction land, including urban and rural built-up land, transportation, and water conservancy facilities accounted for 14.30%. Unused land, including lakes, rivers, tidal flats, swamps, and natural reserve regions, accounted for 8.10% (Fig 1).

### Data sources

Landsat-5 and Landsat-8 satellite images with paths 119 and 39 over the years 1990, 1995, 2000, 2005, 2010, and 2015 were downloaded from the Earth Resources Observation and Science Center, USGS at http://glovis.usgs.gov. The Landsat satellites can provide long-term archive moderate-resolution images, from 15 m to 120 m, of the Earth's land surface and the polar regions [37]. It operates in the visible, near-infrared, short-wave infrared, and thermal infrared spectrum, capturing approximately 400 scenes per day [38]. The 30 m spatial resolution global land cover datasets (Globeland30) for 2000 and 2010 were downloaded from http://www.globallandcover.com/GLC30Download/index.aspx [39]. There are 10 land cover types in Globeland30, with an overall global accuracy of 83% [40]. The land cover type of "artificial surface" in Globeland30 was used in this study. The artificial surface includes various residential

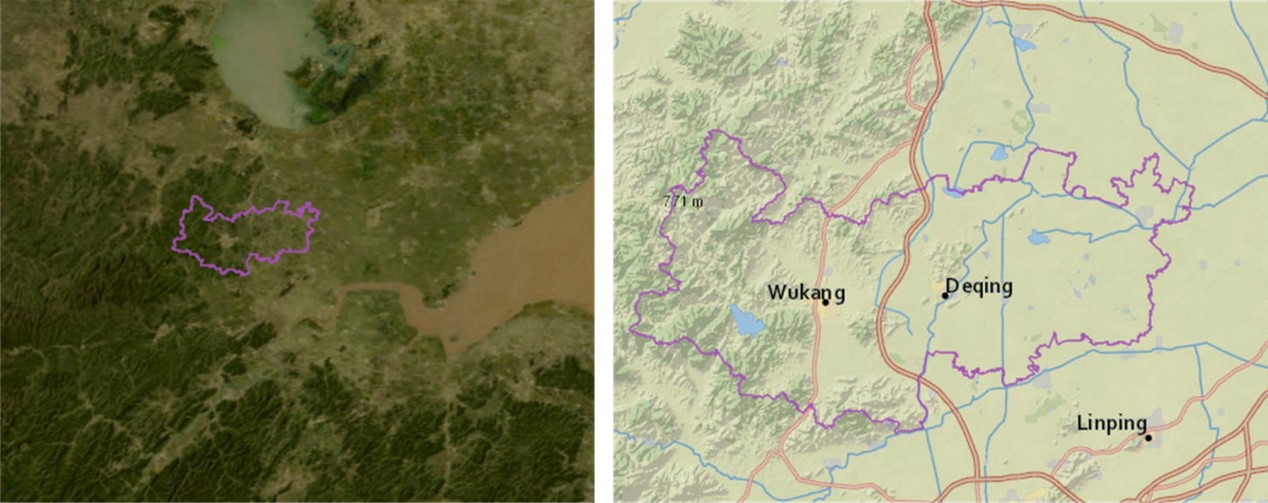

**Fig 1. Study site (extracted from USGS national map viewer).**

areas in cities and townships, industrial and mining lands, and transportation facilities, excluding contiguous green spaces and water bodies within the construction land [41]. The multi-temporal global urban land product for 1990, 1995, 2000, 2005, and 2010, the world's first multi-temporal data set of global impervious surfaces at a 30-m resolution, were downloaded from http://www.geosimulation.cn/GlobalUrbanLand.html [42]. The term "urban land" used in this dataset refers to an impervious surface, i.e., artificial cover and structures, such as pavement, concrete, brick, stone and other man-made impenetrable cover types. It has the same meaning as the land cover of "artificial surfaces" defined in Globeland30. The geographical conditions data for Deqing County were acquired from the local geomatics center. This dataset was produced according to the National Geomatics Standard "Survey Contents and Indicators of Geographical Conditions" (No. GDPJ 01–2013) with 49 land use/land cover types in Deqing County. In addition, basic geographic data such as county and township boundary vector files with a shape file format were acquired from the local Geomatic Center. The township population data for 1990, 1995, 2000, 2005, 2010, and 2015 were collected from the local statistical department of Deqing County. Table 1 lists all the data used in this work.

## Methods

The method for calculating indicator SDG11.3.1 is presented in the SDG indicators Metadata Repository managed by UNDESA (https://unstats.un.org/sdgs/metadata). SDG 11.3.1 is

**Table 1. Spatial and attribute data we used in this work.**

| Data/datasets name | Data Format | Spatial resolution (m) | Temporal resolution |
|---|---|---|---|
| County/township boundaries | Vector | - - | **2015** |
| Landsat 5 TM | Raster | 30 | 1990,1995,2000,2005,2010 |
| Landsat 8 OLI | Raster | 30 | 2015 |
| Globeland30 | Raster | 30 | 2000,2010 |
| Global urban land product | Raster | 30 | 1990,1995,2000,2005,2010 |
| Geographical condition data | Vector | - - | 2015 |
| Township Population | Excel | - - | 1990,1995,2000,2005,2010,2015 |

categorized as a Tier 2 indicator, which means that this indicator is conceptually clear, and the calculation method has been established, but the data is not readily available. Index SDG 11.3.1 is the ratio of the land consumption rate to the population growth rate, which is primarily used to indicate that the urban land use efficiency is globally rapid urbanization [43].

### Population growth rate (PGR)

PGR refers to population growth in a given spatial unit over a period of time. It is mainly used to indicate the number of births and deaths during a period of time as well as the number of people migrating and immigrating.

PGR can be expressed as:

$$\mathrm{PGR} = \frac{\ln(Pop_{t+n}/Pop_t)}{y} \tag{1}$$

where $ln$ is the natural logarithm, and $y$ is the span between the two measurement periods. $Pop_t$ and $Pop_{t+n}$ is the total population within a city in the initial and final year, respectively.

### Land consumption rate (LCR)

LCR is defined as a measure of the percentage of the current total urban land that was newly developed in a given spatial unit over a time span. The land consumption includes: (a) the expansion of built-up areas that can be directly measured; (b) the absolute extent of land that is subject to exploitation by agriculture, forestry, or other economic activities, and (c) the over-intensive exploitation of land used for agriculture and forestry.

LCR can be expressed as:

$$\mathrm{LCR} = \frac{\ln(Urb_{t+n}/Urb_t)}{y} \tag{2}$$

where $Urb_t$ and $Urb_{t+n}$ are the total areal of the urban agglomeration in km$^2$ for the initial and final year, respectively; $ln$ and $y$ are the same meaning as in Eq (1).

### Built-up land extraction

The urban agglomeration areas need to be acquired before calculating the LCR in Eq (2). Based on the SDG 11 monitoring framework (https://unhabitat.org/sdg-goal-11-monitoring-framework), urban agglomeration can be measured as the built-up land. The term "built-up land" has the same meaning as "urban land" in the dataset of Global urban land and "artificial surfaces" in the dataset of Globeland30. Because a 6-year data series of urban lands are needed in this work, no available datasets can meet this requirement. We extracted the built-up lands by integrating the available land cover datasets and manually adjusting them by visually interpreting the satellite images.

**Flowchart to extract the time series built-up lands.** The flowchart for obtaining the built-up lands over the years from 1990 to 2015 with an interval of 5 years is presented in Fig 2.

All the satellite images and remotely sensed products were transformed into GCS_China_ Geodetic_Coordinate_System_2000 to maintain the same coordinate system. Because of the authority with geographical condition data, which can reflect the distribution of national natural resources and national ecological and environmental conditions in geographic space, built-up land related land use/land cover components in geographic condition data, such as buildings, roads, squares, impervious surfaces, and residential areas, were exported and merged

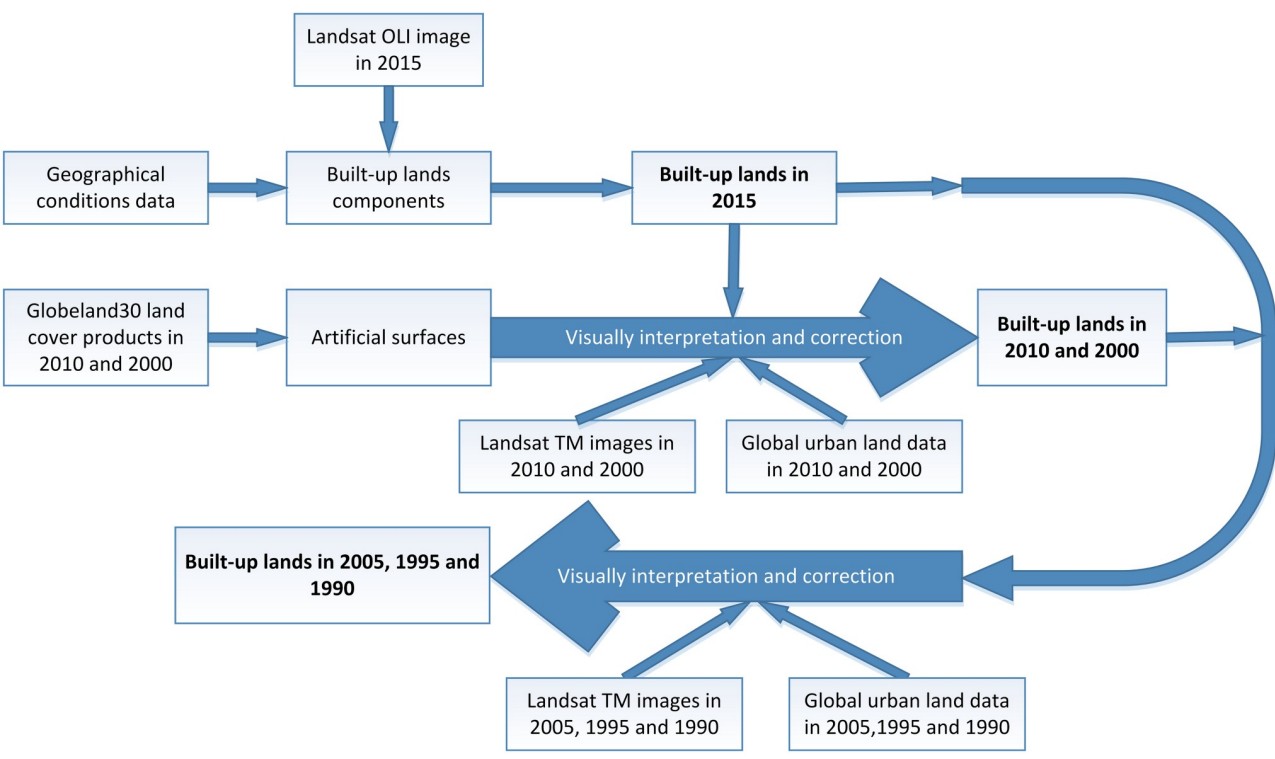

**Fig 2. Flowchart for extracting built-up lands in Deqing County.**

into a single layer, which was regarded as the basis of built-up lands. By aggregating the extracted built-up lands and overlying these on the Landsat OLI false color composition image from 2015, the built-up lands in 2015 were acquired by manually correcting the spatial inconsistency between the built-up lands from the geographical condition data and the actual data from the satellite images, which were identified by visual interpretation. To maintain spatial consistency, the built-up lands from 2015 were used as a reference to obtain the built-up lands for 2010 and 2000. The artificial surfaces from the years 2010 and 2000 were extracted from the Globeland30 dataset. By overlying the Landsat images, the urban land products, and the artificial surfaces from 2010 and 2000, respectively, the built-up lands from 2010 and 2000 were obtained by manually adjusting the classified artificial surfaces to the actual surfaces of the built-up lands reflected from images. With the help of the built-up lands for 2000, 2010, and 2015, the satellite images and urban land products from 2005, 1995, and 1990, we collected the built-up lands for 2005, 1995, and 1990 through manual interpretation and correction. Finally, we obtained built-up lands from 1990 to 2015 with an interval of 5 years to be employed as the urban built-up lands in this study.

**Validation of the extracted built-up lands.** In order to quantitatively verify the built-up land information extracted in this work, a grid of 3 km × 3 km was created on the study site. Ten percent of the grid was randomly selected, and the built-up regions were manually drawn through a visual interpretation of the satellite images from 1990 to 2015 at 5-year intervals. The interpreted built-up lands in the selected grids for each year were used as the actual ground data. The precision, recall, Intersection over Union (IoU) and accuracies were calculated by constructing an error matrix according to the accuracy assessment method by Olofsson et al. in 2014 [44].

### Ratio of land consumption rate to population growth rate (LCRPGR)

LCRPGR is entrusted to quantify the sustainable land use in the face of urban expansion pressures, both demographic and economic. The estimate of the LCRPGR is expressed as

$$LCRPGR = \frac{LCR}{PGR} = \frac{\frac{Ln\left(\frac{Urb_{t+n}}{Urb_t}\right)}{y}}{\frac{Ln(Pop_{t+n}/Pop_t)}{y}} = \frac{Ln(Urb_{t+n}/Urb_t)}{Ln(Pop_{t+n}/Pop_t)} \tag{3}$$

## Results and discussion

### Land expansion in Deqing County

In this work, the expanded built-up lands from two different periods were regarded as land consumption. Based on the method for extraction, we produced built-up lands for the years 1990, 1995, 2000, 2005, 2010, and 2015 in Deqing County. To validate the data from this study, we visually interpreted the built-up lands in 10 randomly selected grids of 3 × 3 km for each year as the reference data. The calculated average precision, recall, Intersection over Union (IoU) and accuracies for each year are presented in Table 2. The mean accuracy is higher than 94%, which means that our extracted built-up lands achieved good accuracy.

Based on the extracted built-up lands, we obtained the urban expansion from 1990 to 2015, with at 5-year intervals in Deqing County (Fig 3). During the period from 1990 to 2015, the built-up lands presented a phenomenon of urban sprawl, increasing by 129 km² from 15 km² in 1990 to 144 km² in 2015 (Fig 4). Furthermore, the built-up lands rapidly increased from 2000 to 2005, which can be clearly seen from the largest gradient during this period compared with the others.

There were 11 townships in Deqing County over the time span of this investigation. They generally presented an increasing trend in the areas of built-up lands from 1990 to 2015 at 5-year intervals. From 2000 to 2005, nearly all the townships demonstrated a peak increase in built-up lands. This demonstrated a process of rapid urban development and construction during this time span. Some townships, such as Wukang, the county center, and Xinshi, a town with heritage cultures, expanded considerably, while townships of Fatou and Moganshan presented a small trend of increase in built-up lands (Fig 5), indicating that these townships had not suffered much from the process of urbanization.

### Population growth

The employed population number is the population registered in the household from the local statistical department. It presented an increasing trend from 1990 to 2015 throughout Deqing County (Fig 6), increasing by 38,000 over the period. Compared with the continuous increase of built-up lands in the county centers and surrounding townships, the demographic changes in townships shifted. From 1990–2000 and 2005–2015, the population in Wukang, the county center, has been increasing (Fig 7). The population changes from 1995 to 2005 were

**Table 2. Accuracies of the extracted built-up lands in the study site.**

| Name | 1990 | 1995 | 2000 | 2005 | 2010 | 2015 |
|------|------|------|------|------|------|------|
| Precision | 85.60 | 62.10 | 71.56 | 67.36 | 73.24 | 80.67 |
| Recall | 79.90 | 91.78 | 81.97 | 74.55 | 82.48 | 85.96 |
| IoU | 70.43 | 58.83 | 61.83 | 54.77 | 63.38 | 71.27 |
| Accuracy | 92.52 | 94.11 | 95.32 | 94.60 | 95.29 | 97.47 |

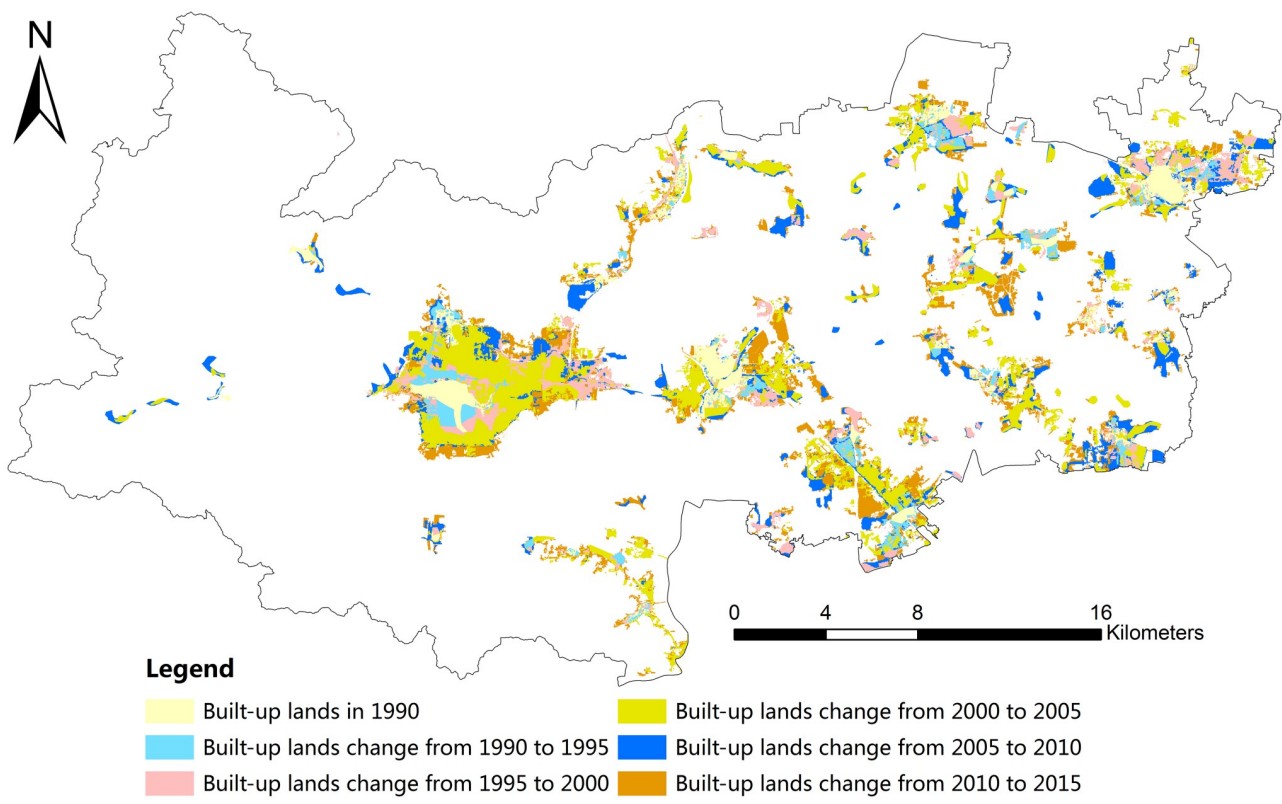

**Fig 3. Expansion of built-up lands in Deqing County from 1990 to 2015.**

predominantly caused by the adjustment of local administrative divisions. In 1995, the county center was moved from township Qianyuan to Wukang, as a result, many people moved from Qianyuan to Wukang From 1995 to 2000. From 2000 to 2005, a region currently named Fuxi was adjusted to outside the jurisdiction of Wukang, decreasing the population. Generally, people in townships moved to the county center, which caused the rapid urbanization in Wukang.

## Ratio of LCR to PGR

From Fig 7, we can see that there were large shifts in the township population caused by the local policies. The aim of SDG 11.3.1 is to identify land use efficiency during the process of

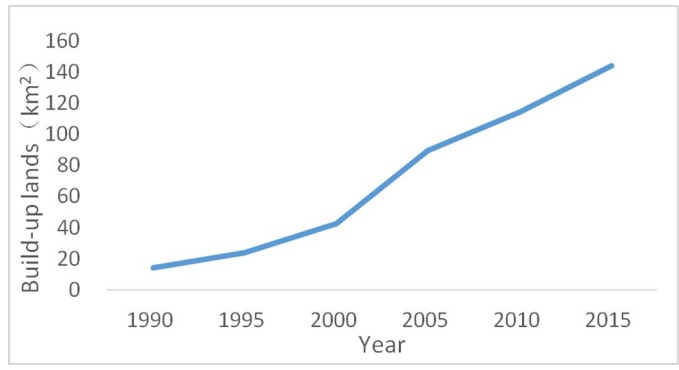

**Fig 4. Change in built-up lands in Deqing County from 1990 to 2015.**

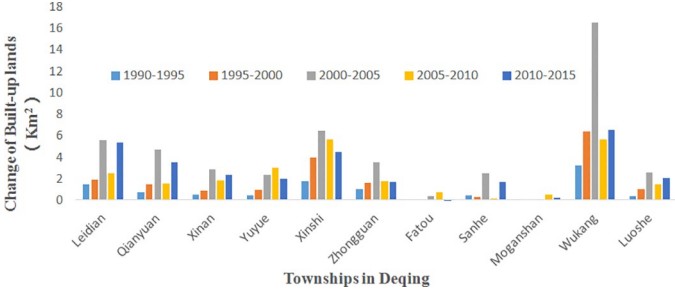

**Fig 5. Built-up land change in Deqing townships from 1990 to 2015.**

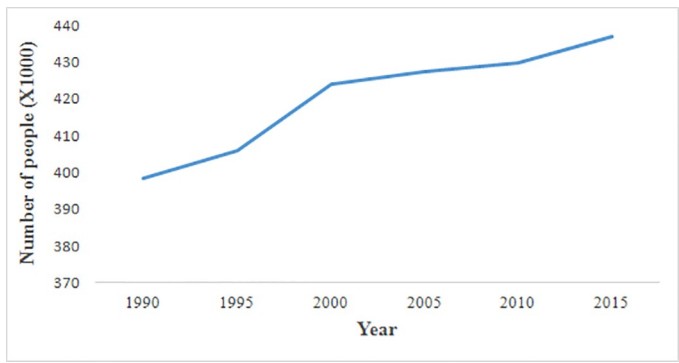

**Fig 6. Change in population in Deqing County from 1990 to 2015.**

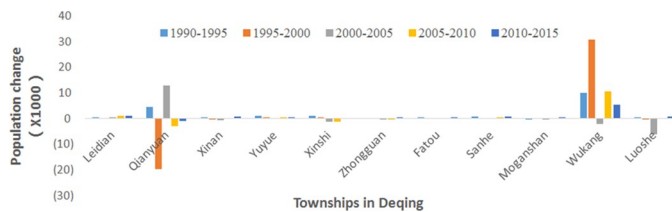

**Fig 7. Variation in population in townships of Deqing from 1990 to 2015.**

urbanization; therefore, we calculated the LCRPGR values at 5-years intervals for Wukang, the county center of Deqing, which has suffered from rapid urbanization from 1990 to 2015 (Table 3). Because of the decrease in population from 2000 to 2005, the LCRPGR has a negative value. The remaining LCRPGR values were 4.28, 1.88, 2.23, and 3.81 for the other time

**Table 3. PGR, LCR, and LCRPGR in township of Wukang, the county center.**

| | Indicators | 1990–1995 | 1995–2000 | 2000–2005 | 2005–2010 | 2010–2015 |
|---|---|---|---|---|---|---|
| Wukang | PGR | 0.0286 | 0.0695 | -0.0007 | 0.0155 | 0.0089 |
| | LCR | 0.1225 | 0.1304 | 0.1610 | 0.0346 | 0.0338 |
| | LCRPGR | 4.28 | 1.88 | -234.95 | 2.23 | 3.81 |
| | PGR | 0.0974 | | | 0.1219 | |
| | LCR | 0.4139 | | | 0.3419 | |
| | LCRPGR | 4.25 | | | 2.81 | |

spans. For the long-term land use efficiency, the LCRPGR from 2005 to 2015 is smaller than that from 1990 to 2005. This indicates that the land use efficiency increased from 1990 to 2015 over the long term. Based on the LCRPGR results, with an interval of 5 years or more than 10 years, the average speed for built-up land construction is approximately 3 times faster than the population growth, which is more than 1.5, which is regarded as the average ratio of LCR to PGR in the last 20 years worldwide (UN Habitat, 2018). This result indicates that Wukang is less dense than the average cities around the world. This phenomenon is primarily because our study site is a county center with only small urban areas. However, in the long term of 1990–2005 and 2005–2015, the county center has become denser as it grows, which will positively affect the sustainability of urban development.

It should be noted that the negative LCRPGR value from 2000 to 2005 should be excluded to calculate the mean LCRPGR from 1990 to 2015; otherwise, the mean LCRPGR value will not be representative of the relationship between the land consumption rate and population growth rate. This means that if either the LCR or PGR is negative, the LCRPGR must have a negative value, especially when there is a very small decrease in population from the initial year to the final year. When the land consumption is a positive value, the calculated ratio between LCR and PGR will be substantially negative, as demonstrated by the ratio calculated for Wukang from 2000 to 2005. If this negative value is included in the calculation of the average LCRPGR, the mean of LCRPGR will be much smaller than that computed statistically without the negative value; therefore, the ratio of LCR to PGR cannot be used as an indicator of land use efficiency.

## Discussion

A technical flowchart was designed in this study to obtain spatially consistent built-up lands for 1990 to 2015 by integrating four different types of data. By comparing the results with the investigated construction land from the local government, the extracted built-up lands in this work achieved good accuracy. This indicates that high-resolution satellite images, geographical condition data, and land cover products are useful data resources for extracting built-up lands. This work provides an approach to obtain time-series built-up land data with a good accuracy to calculate the land consumption rate and estimate the urban land use efficiency. Because of the high accuracy with the geographical conditions data reflecting the true land cover/land use status, it played a key role in obtaining the built-up lands in different periods. Because of the uncertainty in information extraction from the satellite images, more time was required to integrate the data from different data sources and manually edit the data to ensure that the finally extracted built-up lands for different years can be spatially consistent. As a result, the extracted built-up lands achieved good accuracy, while it was a time-consuming job.

The population data we used were the census of the registered household population in a spatial unit of the townships. Due to rapid urbanization, population mobility has become a common phenomenon, so it is difficult to obtain the permanent population of each township that meets the requirements of the time series data. Additionally, according to the monitoring framework (UN habitat, 2018), many villages were not included in the calculation of built-up lands because of the vast distance from the county center, but the registered household population is the entire population in a region of an administrative area, so the population used in this study was not completely consistent with the built-up lands. Therefore, population and the consistency between the population and the geometry of the built-up lands is a key issue in the calculation of SDG 11.3.1, which is also our focus in further studies.

The SDG-defined indicator is used for the urban development assessment. The study area is a county center with a much larger rural area than a city. The LCRPGR values for the study

site were larger than the average value, indicating that the urban land use efficiency in the country center is lower than the average value of cities worldwide. Compared with small- or medium-sized cities, the study site contains substantial rural areas; it is normal to have relatively less dense construction land areas. This is the primary reason the study site has larger LCRPGR values. From a long-term perspective, it is advantageous that urban land use becomes denser as it grows, which will be beneficial for realizing SDG 11 in the study site.

During the calculation of indicator SDG 11.3.1, we determined negative LCRPGR values that were caused by a population decrease during the study period. The negative LCRPGR value indicates that the method given by the UN to calculate the SDG 11.3.1 is specifically for cities suffering from rapid urbanization with positive values for both population growth and land consumption. If there is one negative value in either LCR or PGR, the calculated LCRPGR cannot be included in the computation of the mean LCRPGR value in a region with several cities or a long time span consisting of several short time spans. This study is only concerned with one case; we need more work to test the method of indicator SDG 11.3.1 given by the metadata by investigating more cities in China and worldwide.

## Conclusions

Urbanization is a current continuing trend worldwide, especially in Asia, Africa, and South America. SDG 11.3.1 was designed to calculate the efficiency of urban land use as a consequence of population growth. Conclusions from a case study in Deqing, a county in Huzhou of Zhejiang Province, can be drawn as follows:

Satellite imagery, land use, and land cover products can provide continuous and stable data sources to calculate the land consumption rate, population growth rate, and the corresponding land use efficiency in a time series format, although many satellite image processing techniques are required. The designed data processing flowchart can be helpful for others to extract built-up lands.

By combining spatial data with statistical data, the SDG11.3.1 indicator can be used to quantitatively analyze the condition of urban land supply and urban land use efficiency as the urban population grows. Generally, the land consumption in Deqing County center has been approximately 3 times faster than the population growth from to 1990–2015, while in the long term, urban land use becomes denser as it grows, which means that land use inefficiency is changing rapidly in the process of urbanization in Wukang, the county center of Deqing. This will have a positive effect on the urban sustainability of the study area.

## Supporting information

**S1 Data.**
(XLS)

## Acknowledgments

We sincerely thank the Earth Resources Observation and Science Center, USGS for providing the Landsat images and the relevant institutions in Deqing County, such as the city administrative office, local statistical department, local survey department, and local city planning department for their support in the local statistical data provision.

## Author Contributions

**Conceptualization:** Guoyin Cai.

**Data curation:** Jinxi Zhang.

**Formal analysis:** Guoyin Cai.

**Investigation:** Jinxi Zhang, Chaopeng Li.

**Methodology:** Guoyin Cai.

**Project administration:** Shu Peng.

**Resources:** Mingyi Du.

**Validation:** Jinxi Zhang, Chaopeng Li.

**Writing – original draft:** Guoyin Cai.

**Writing – review & editing:** Guoyin Cai.

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
