## [Decision Letter · Decision Letter 0]

13 Sep 2020

PONE-D-20-23738

Identification of Urban Land Use Efficiency by Indicator-SDG 11.3.1

PLOS ONE

Dear Dr. Cai,

Thank you for submitting your manuscript to PLOS ONE. After careful consideration, we feel that it has merit but does not fully meet PLOS ONE’s publication criteria as it currently stands. Therefore, we invite you to submit a revised version of the manuscript that addresses the points raised during the review process.

We look forward to receiving your revised manuscript.

Kind regards,

Changshan Wu

Academic Editor

PLOS ONE

Journal Requirements:

2. Thank you for submitting the above manuscript to PLOS ONE. During our internal evaluation of the manuscript, we found significant text overlap between your submission and the following previously published works, some of which you are an author.

int-arch-photogramm-remote-sens-spatial-inf-sci.net/XLII-3-W10/1073/2020/isprs-archives-XLII-3-W10-1073-2020.pdf

Please revise the manuscript to rephrase the duplicated text, cite your sources, and provide details as to how the current manuscript advances on previous work. Please note that further consideration is dependent on the submission of a manuscript that addresses these concerns about the overlap in text with published work.

6. We note that Figure 1 and 3 in your submission contain map images which may be copyrighted. All PLOS content is published under the Creative Commons Attribution License (CC BY 4.0), which means that the manuscript, images, and Supporting Information files will be freely available online, and any third party is permitted to access, download, copy, distribute, and use these materials in any way, even commercially, with proper attribution. For these reasons, we cannot publish previously copyrighted maps or satellite images created using proprietary data, such as Google software (Google Maps, Street View, and Earth). For more information, see our copyright guidelines: http://journals.plos.org/plosone/s/licenses-and-copyright.

6.1.    You may seek permission from the original copyright holder of Figure 1 and 3 to publish the content specifically under the CC BY 4.0 license. 

6.2.    If you are unable to obtain permission from the original copyright holder to publish these figures under the CC BY 4.0 license or if the copyright holder’s requirements are incompatible with the CC BY 4.0 license, please either i) remove the figure or ii) supply a replacement figure that complies with the CC BY 4.0 license. Please check copyright information on all replacement figures and update the figure caption with source information. If applicable, please specify in the figure caption text when a figure is similar but not identical to the original image and is therefore for illustrative purposes only.

Reviewers' comments:

Reviewer's Responses to Questions

**Comments to the Author**

1. Is the manuscript technically sound, and do the data support the conclusions?

Reviewer #1: Yes

Reviewer #2: Partly

2. Has the statistical analysis been performed appropriately and rigorously? 

Reviewer #1: Yes

Reviewer #2: Yes

3. Have the authors made all data underlying the findings in their manuscript fully available?

Reviewer #1: Yes

Reviewer #2: No

4. Is the manuscript presented in an intelligible fashion and written in standard English?

Reviewer #1: Yes

Reviewer #2: Yes

5. Review Comments to the Author

Reviewer #1: This is an interesting work, and it is well written. I recommend a minor revision for the manuscript.

1. Introduction. This section is well organized, however, the scientific question you are trying to address is not very clear. I feel it is an interesting work, but I do not feel your work is very important. Please highlight your contributions.

2. Study area. Since you are talking about land use efficiency, please provide more information about the land use in the study area.

3. Results and discussion. You did a good job in describing your research results. However, limited discussion I can see here, this part needs to be improved.

4. Conclusions. The limitation of your work and possible future search directions should also be pointed out here.

Reviewer #2: The study adopts the SDG-defined LCGPRG ratio to extract the UN SDG11.3.1 indicator of land use efficiency in a county level in China, and examines its multi-period variations in a 5-year interval. The study design is easy to follow, but the manuscript needs improvement in writing for readers to better understand the context. Major concerns include:

- Introduction: More details about SDG indicators are needed. The SDG indicators are established to assess our environment in global scale. Specifically about urban, Please provide more background on SDG11. There have been quite a number of global urban settlement databases (both impervious surface and population grid) available online. This manuscript only mentions the GHSL and GPW. In Page9 (please note this is the page number of the PDF package, not the manuscript), “…Rozhenkova et al. (2019) found that the current existing databases are insufficient for the purpose of large-scale comparative analysis….” What are these databases? Why are they not sufficient? In Page 10, “…localization is a key problem of SDG indicators…” What does the “localization” mean? Actually, both GHSL and GPW are km-scale global layers. The GHSL is extracted from Landsat imagery per my understanding. What is the rationale that the authors decide to extract their own built-up layers from Landsat imagery instead of using these global products? I believe the 5-year data series is a good justification.

- Study Area: please describe all Data Sets used in the study. It utilizes Landsat imagery, official land use maps from local government, Globelland30 product, and more probably. I guess that the study also utilized the GHSL but could not find any description here. All the information has been scattered in the Methods and Results sections. More details are needed. For example, what is the definition of “artificial surfaces” in Globeland30? It seems that two official land use maps (2010 and 2015) have been utilized in this study (page 14). However, only the 2015 one is mentioned without much details. What does the authors define the built-up land (in other words, how to match the built-up lands in different sources)?

- Methods: I would also suggest that the “Built-up lands extraction” belongs to the Methods section because the authors have made intensive image analysis. Also more details are need in this subsection: what does the “geographical conditions” mean? What are the criteria for manual adjustment (e.g. minimum polygon)? What are the “global urban guilt-up data” in the flowchart in Fig.2? Accuracy assessment is a necessary component but is missing.

- Methods: “Land consumption Rate” in Page 13, does the land consumption include agricultural and forest lands at all? It will make a big difference of the SGD11.3.1 indicator.

- Results: Table 1 in Page 14 is not really accuracy assessment. What are the causal factors of the difference between the investigated and extracted built-up lands? Especially in 2010.

- “Population Growth” in Page 15-16: Here I have the biggest concern of this study - the validity of PGR. The authors mention that “….There is a decrease in population from 1995 to 2000 (Fig6)…” However, the reasons of this decrease are poorly described in the context. What kind of “local policies”? How do these policies affect the “correspondingly adjustment”? Specifically for Wukang, I could not see the 2000-2005 decreasing trend in Fig.7, which is related to the negative LCRPGR value that is an important piece of conclusion. Actually, Fig.7 shows dramatic 1995-2000 decrease of two townships (Qianyuan and Yuyue) and the similarly dramatic increase in 2000-2005. Note their long-term overall population changes quite low in both townships. What are the causes of these changes? Are these short-term changes meaningful to LCGPRG?

- Table 2 could serve as a final deliverable of this study. However, why does it only include Wukang? What about other townships? How about the whole county? How about the 1995-2015 long-term land use efficiency?

- Discussion: It seems to me that the SDG-defined indicator is for urban development assessment. As shown in Fig.1, the study area is actually not urban land. Even for the Wukang township, there is still a much larger rural area than the city. I would suggest the authors discuss the suitability of this SDG11 indicator in rural areas. It could be valuable information for urban SDG community.

6. PLOS authors have the option to publish the peer review history of their article (what does this mean?). If published, this will include your full peer review and any attached files.

Reviewer #1: No

Reviewer #2: No

---

## [Author Response · Author response to Decision Letter 0]

31 Oct 2020

Dear reviewers, 

Thanks for your comments. we have revised our manuscript based on your valuable coments one by one. Please see detail in the document of response to reviewers and the revised version.

---

## [Decision Letter · Decision Letter 1]

8 Dec 2020

Identification of Urban Land Use Efficiency by Indicator-SDG 11.3.1

PONE-D-20-23738R1

Dear Dr. Cai,

We’re pleased to inform you that your manuscript has been judged scientifically suitable for publication and will be formally accepted for publication once it meets all outstanding technical requirements.

Kind regards,

Changshan Wu

Academic Editor

PLOS ONE

Additional Editor Comments (optional):

Reviewers' comments:

Reviewer's Responses to Questions

**Comments to the Author**

1. If the authors have adequately addressed your comments raised in a previous round of review and you feel that this manuscript is now acceptable for publication, you may indicate that here to bypass the “Comments to the Author” section, enter your conflict of interest statement in the “Confidential to Editor” section, and submit your "Accept" recommendation.

Reviewer #2: All comments have been addressed

2. Is the manuscript technically sound, and do the data support the conclusions?

Reviewer #2: Yes

3. Has the statistical analysis been performed appropriately and rigorously? 

Reviewer #2: Yes

4. Have the authors made all data underlying the findings in their manuscript fully available?

Reviewer #2: Yes

5. Is the manuscript presented in an intelligible fashion and written in standard English?

Reviewer #2: Yes

6. Review Comments to the Author

Reviewer #2: I would like to thank the authors for their tremendous efforts on revising and polishing the manuscript. All of my review comments have been properly addressed.

7. PLOS authors have the option to publish the peer review history of their article (what does this mean?). If published, this will include your full peer review and any attached files.

Reviewer #2: No

---

## [Editor Report · Acceptance letter]

15 Dec 2020

PONE-D-20-23738R1 

Identification of Urban Land Use Efficiency by Indicator-SDG 11.3.1 

Dear Dr. Cai:

I'm pleased to inform you that your manuscript has been deemed suitable for publication in PLOS ONE. Congratulations! Your manuscript is now with our production department. 

Kind regards, 

on behalf of

Dr. Changshan Wu 

Academic Editor

PLOS ONE